# Health Information Use and Trust: The Role of Health Literacy and Patient Activation in a Multilingual European Region

**DOI:** 10.3390/ijerph22040570

**Published:** 2025-04-05

**Authors:** Christian J. Wiedermann, Verena Barbieri, Stefano Lombardo, Timon Gärtner, Patrick Rina, Klaus Eisendle, Giuliano Piccoliori, Adolf Engl, Dietmar Ausserhofer

**Affiliations:** 1Institute of General Practice and Public Health, Claudiana College of Health Professions, 39100 Bolzano, Italy; 2Provincial Institute for Statistics of the Autonomous Province of Bolzano, South Tyrol (ASTAT), 39100 Bolzano, Italy; 3Directorate, Claudiana College of Health Professions, 39100 Bolzano, Italy; 4Claudiana Research, Claudiana College of Health Professions, 39100 Bolzano, Italy

**Keywords:** health information-seeking behaviour, trust in health sources, health literacy, patient activation, digital health literacy

## Abstract

The acquisition of reliable health information plays a pivotal role in shaping informed decision-making and health-related behaviours. This investigation examined the factors influencing health information use and trust in health information sources among the adult population in South Tyrol, Italy, employing a population-based cross-sectional survey (*n* = 2090). Descriptive analyses revealed sociodemographic disparities, with younger individuals and those with higher educational attainment demonstrating increased engagement with digital sources, while older adults and those with lower educational levels exhibited a greater reliance on traditional media and healthcare professionals. Correlation analyses showed that elevated health literacy (HLS-EU-Q16) was linked to enhanced engagement with structured and professional health sources, whereas higher patient activation (PAM-10) exhibited a negative correlation with a dependence on healthcare professionals, indicating that more activated individuals are less reliant on medical consultations for health information. Individuals reporting a better health status were less inclined to use health information sources, such as media, healthcare professionals, or the internet, as opposed to relying on personal knowledge or experience. Ordinal regression models further identified age, education, and linguistic background as crucial predictors of health information use and trust in sources. These findings highlight the influence of health literacy and patient activation on information engagement and trust and emphasise the need for tailored public health initiatives to ensure equitable access to reliable health information across diverse demographic groups.

## 1. Introduction

Access to reliable health information is a fundamental determinant of informed decision-making and health-related behaviours [1,2]. In the contemporary digital era, individuals obtain health information from a wide variety of sources, including personal healthcare providers such as general practitioners, nurses, and health coaches, professional experts like nutritionists, scientists, and academics, official government websites (e.g., WHO, CDC, and NHS), and digital channels including search engines, health-related websites, scientific news outlets, blogs, podcasts, and social media platforms such as Facebook and WhatsApp. Interpersonal networks (e.g., family, friends, and colleagues) and traditional media sources like books, newspapers, and television also continue to play a role in shaping how health information is accessed and shared [3,4,5].

However, the selection of health information sources and the trust placed in them vary across different demographic and sociocultural groups, influencing both health literacy and patient activation [6,7]. Throughout this study, the term “reliance” refers specifically to the self-reported frequency of using a particular health information source, as assessed by Likert-scale responses. It does not imply trust or habitual dependence on the source for decision-making unless otherwise stated. This usage aligns with prior research distinguishing reliance from trust and active seeking [8,9]. Trust is defined as a willingness to accept vulnerability based on positive expectations of the intentions or behaviour of a source, even in the absence of direct control or the ability to verify the information provided [10,11]. Use is a general term encompassing both active and passive engagement with sources. Understanding the determinants of health information use and trust is crucial for tailoring effective public health communication strategies [12].

Health information use is influenced by multiple sociodemographic factors, including age, sex, education level, and cultural group membership [13,14]. Research has demonstrated that younger individuals and those with higher education are more inclined to engage with digital health sources, whereas older adults tend to rely on traditional media and healthcare professionals [15]. Gender disparities have also been observed, with women generally exhibiting higher engagement in health information-related activities than men [7]. In bilingual and multilingual regions, linguistic group membership may further influence access to and preference for different health information sources [15]. However, empirical evidence regarding these determinants in South Tyrol, a linguistically diverse region in Italy, remains limited.

Health literacy, defined as the ability to access, comprehend, and utilise health information effectively, plays a crucial role in self-care and decision-making [6]. The Patient Activation Measure (PAM-10) assesses an individual’s knowledge, skills, and confidence in managing their health, with higher activation levels associated with improved health outcomes and self-care behaviours [16]. A substantial body of research suggests that digital health information seekers—individuals who actively search for health information online—may demonstrate higher levels of health literacy and patient activation as they are more engaged in proactive health management [7]. However, the applicability of these trends in South Tyrol, where both linguistic and regional differences may influence access to digital health information, remains unclear.

Trust is a key determinant of whether individuals accept and act on health information or not. Studies indicate that healthcare professionals (general practitioners, specialists, pharmacists, and nurses) are among the most trusted sources, whereas trust in internet-based sources and social media is comparatively low [15]. Nevertheless, some individuals, particularly those with lower health literacy, may overly rely on online information and exhibit lower trust in traditional medical sources [7,17]. It is also unclear whether trust in traditional media (newspapers, TV, and books) correlates positively with health literacy and patient activation levels.

This study aimed to explore the determinants of health information use and trust in health information sources among adults in South Tyrol. To systematically investigate the relationship between demographic and health-related factors with health information behaviours and trust in information sources, we applied the conceptual framework outlined in Figure 1. This framework illustrates the independent variables considered in this analysis, the dependent variables representing information-seeking behaviour and trust, and the categorization of different types of health information sources. Specifically, we addressed the following research questions:What sociodemographic factors influence health information use and trust in health sources?How do health literacy and patient activation relate to health information use?What factors predict trust in health information sources, and how does trust influence engagement with information sources?

By addressing these questions, this study sought to provide insights into how individuals in South Tyrol navigate health information, with implications for public health communication, digital health strategies, and healthcare professional engagement.

## 2. Materials and Methods

### 2.1. Study Design, Setting, and Sample

From 1 March to 30 May 2024, a comprehensive, multi-sectoral population survey was conducted in the Autonomous Province of Bolzano, South Tyrol. This research initiative was a collaborative endeavour between the Provincial Institute of Statistics (ASTAT; Istituto Provinciale di Statistica-Landesinstitut für Statistik) and the Institute of General Practice and Public Health.

The Autonomous Province of Bolzano, also known as South Tyrol, is situated in the Trentino–Alto Adige region of Italy, bordering Austria. With a total population of 535,000, the linguistic composition comprises approximately 70% of German speakers, 25% of Italian speakers, and 5% of speakers of Ladin and other languages. The survey’s target demographics encompassed approximately 400,000 residents of South Tyrol aged 18 years and above. This study employed stratified probabilistic sampling methodology. ASTAT conducted a random selection of adults aged 18 years and older from the province-wide register of current residents. This selection was stratified according to age groups (18–34, 35–54, and 55 and above), gender (male and female), citizenship status (Italian or other), and place of residence (municipalities). A sample of 4000 individuals was drawn to achieve an appropriate level of precision considering the distribution and variation across the strata.

### 2.2. Participant Survey

ASTAT and the Institute of General Practice and Public Health jointly created the participant questionnaire. The German and Italian versions provided by ASTAT underwent a linguistic equivalence review by researchers at the Institute for General Practice and Public Health. The final questionnaire comprised 91 items, covering sociodemographic data, health information use, trust in health sources, health literacy (HLS-EU-Q16), patient activation (PAM-10), and self-reported health status. Not all items are analysed in the present study; selected sections will be used in future publications. A machine-translated English version of the full questionnaire using DeepL is provided in Appendix A. The translation was subsequently reviewed and manually corrected by a bilingual researcher to ensure linguistic accuracy and contextual consistency.

#### 2.2.1. Health Information Use and Trust

The evaluation included ten common sources of health-related information: (1) layperson medical discussions (being asked for advice), (2) magazines and newspapers, (3) television and radio broadcasts, (4) friends and acquaintances, (5) healthcare professionals, (6) educational courses, (7) the medical literature, (8) general online searches, (9) targeted searches in electronic databases, and (10) online forums. To enable a direct comparison of the results with those from a decade ago, the same survey questions as those used by Ausserhofer et al. [15] were selected. These self-developed items, adapted from existing instruments but without formal validation, were assessed using a 4-point Likert scale ranging from 1 (“regularly”) to 4 (“never”). The survey item was phrased to capture both active engagement with information (e.g., targeted searches) and passive exposure (e.g., encountering information by chance).

Trust in various sources of health information was assessed utilising the following inquiry: “How much do you trust the following sources for health information?” The evaluated sources comprised (1) specialists in outpatient clinics or hospitals, (2) family doctors, (3) personal feelings or experiences, (4) pharmacists, (5) nurses, (6) information from books, (7) advice from friends or relatives, and (8) information from the internet. Responses were recorded on a 4-point Likert scale ranging from 1 (“very”) to 4 (“not at all”).

#### 2.2.2. HLS-EU-Q16

Health literacy was assessed using the European 16-item Health Literacy Survey (HLS-EU-Q16), which measures an individual’s ability to access, understand, appraise, and apply health-related information in healthcare, disease prevention, and health promotion contexts [18]. The HLS-EU-Q16 is a validated short version of the original 47-item HLS-EU questionnaire (HLS-EU-Q47) [19], retaining 16 key items while maintaining reliability and validity.

The questionnaire was administered in German and Italian using validated translations that demonstrated strong psychometric properties, including high internal consistency and construct validity [18,20,21]. Each item was rated on a 4-point Likert scale: (1) very difficult, (2) fairly difficult, (3) fairly easy, and (4) very easy. Following the HLS-EU-Q16 manual, the response option ‘don’t know’ was treated as a missing value and excluded from scoring. Participants were only included in the analysis if they provided at least 13 valid responses. Responses were summed to a raw score (0–16), which was then transformed into a standardised health literacy index ranging from 0 (lowest) to 50 (highest), following the established HLS-EU-Q16 scoring procedure. Categories were defined as follows: inadequate (0–25), problematic (26–33), and adequate (34–50). This transformation allows for comparability with other European HLS-EU studies. Alternative classifications based on raw sum scores exist but were not applied in this study.

#### 2.2.3. Patient Activation Measure 10 (PAM-10)

Patient activation was assessed using the 10-item Patient Activation Measure (PAM-10), which evaluates individuals’ knowledge, skills, and confidence in managing their health [22].

The PAM-10 is a shortened version of the original 13-item PAM (PAM-13), which retains 10 of the original items while maintaining reliability and validity. As no specifically validated PAM-10 versions were available in Italian or German, the corresponding 10 items from the validated PAM-13 versions in these languages were used. These validated translations have demonstrated good psychometric properties, including high internal consistency and construct validity [16,23].

Each item was rated on a 5-point Likert scale: (1) strongly disagree, (2) disagree, (3) neutral, (4) agree, and (5) strongly agree. Raw scores were transformed into a standardised activation score ranging from 0 (lowest activation) to 100 (highest activation) following the established PAM scoring methodology. The PAM score was categorised into four levels to provide a structured interpretation of patient activation [24]:Level 1 (≤47.0): Disengaged and overwhelmed.Level 2 (47.1–55.1): Becoming aware but still struggling.Level 3 (55.2–72.4): Taking action.Level 4 (≥72.5): Maintaining behaviours and pushing further.

#### 2.2.4. Demographic and Health Characteristics

The survey collected data on the respondents’ demographic characteristics, including birth year, gender (male/female), native language (German/Italian/Ladin or Others), citizenship (Italian/other), educational level (below school/high school or higher), community and region of origin (rural/urban), and living situation (alone/with spouse, family member, or with parents or children). Health-related variables included self-reported health status on a scale of 1–100.

ASTAT dispatched letters to a random selection of potential participants, inviting them to voluntarily participate in this study. Respondents were given the option to complete the survey independently or with the assistance of a family member. The survey was completed online or via telephone interviews conducted by the ASTAT collaborators. A follow-up letter was sent one month after the initial communication to remind individuals about the study and encourage their participation. The survey platform, LimeSurvey [25], was used to create an online questionnaire.

### 2.3. Statistical Analysis

Health literacy (HLS-EU-Q16) scores were categorised into inadequate, problematic, and adequate levels following the standard classification criteria. Patient activation (PAM-10) was categorised into four levels, with the highest proportion of respondents in Level 2 indicating that they were becoming aware but still struggling with health-related self-management. Self-reported health status was categorised based on the WHO clinical thresholds, ranging from poor to excellent health. These weighted distributions provided a representative overview of the study population.

Only fully completed questionnaires were included in the statistical analyses. Descriptive statistics were used to describe the measured variables. To adjust for non-participation bias (i.e., differences between respondents and non-respondents) and ensure that the sample was representative of the target population, weighted descriptive statistics were calculated using post-stratification weights using iterative proportional fitting (raking) to align the sample distributions with population-level margins for age group, gender, citizenship, and municipality of residence, as provided by ASTAT. The weights were scaled to the sample size. Weighted medians and interquartile ranges were computed for continuous variables. Weighted proportions and 95% confidence intervals (CIs) were reported for categorical variables. No imputation was performed; only fully completed questionnaires were analysed. Notably, 95% confidence intervals for categorical variables were calculated using the Wald method.

To assess differences in health information source use across age groups, one-way analysis of variance (ANOVA) was conducted. The dependent variable was age (continuous), and the independent variable was the frequency of using each health information source (four levels: regular, sometimes, seldom, and never). The homogeneity of variances was tested using Levene’s test, and when violated, the robustness of the ANOVA with large sample sizes was considered. Post hoc pairwise comparisons were performed using Tukey’s honest significant difference test with adjusted *p*-values for multiple comparisons. Effect sizes were estimated using omega squared (ω^2^) to quantify the difference magnitude, interpreted as very small (≤0.01), small (0.01–0.06), moderate (0.06–0.14), and large (≥0.14) [26].

Sample size estimation was based on general guidelines for population-based survey research and recommendations for detecting small effect sizes (ω^2^ = 0.02) in subgroup analyses. Prior survey research has suggested that a sample size of approximately 1000 respondents is sufficient to ensure representativeness and detect small differences in health behaviour studies [27,28]. Given the planned subgroup analyses, the final target sample size was increased to 2000 participants to account for stratification and potential non-response adjustments.

Weighted Spearman’s rank correlation coefficients (ρ) were computed to examine the relationships between patient activation (PAM-10), health literacy (HLS-EU-Q16), and self-reported health status (0–100 scale) with the use of ten health information sources: newspapers/magazines, TV/radio, friends/acquaintances, healthcare professionals, events/courses, the specialist literature, incidental exposure to health information online, targeted internet searches, internet forums, and social media.

Ordinal regression models were employed to investigate the predictors of health information use and trust in health information sources, adhering to established recommendations for ordinal regression in public health research [29]. The analyses examined the influence of age, sex, education level, geographic region, native language, health literacy (HLS-EU-Q16), patient activation (PAM-10), and subjective health status on both outcomes. Each source was analysed separately to account for potential differences in predictor effects. Given the ordered nature of the dependent variables, ordinal regression with a cumulative logit link function was used. All predefined predictors were retained in the models, including geographic region and native language, owing to their relevance to South Tyrol.

Model performance was evaluated using Akaike Information Criterion (AIC), Bayesian Information Criterion (BIC), deviance, and Pearson goodness-of-fit tests. Odds ratios (ORs) and 95% confidence intervals (CIs) were derived from the regression coefficients. No formal collinearity diagnostics were conducted; however, given the exploratory nature of this study, collinearity was not anticipated to substantially impact the interpretation. Predictors were selected based on prior research and theoretical frameworks, rather than data-driven selection.

Findings were categorised into traditional media, personal sources, literature-based sources, online sources for health information use and healthcare professionals, personal/social trust, and educational/media sources for trust analyses.

Analyses were conducted using Jeffreys’ Amazing Statistics Program (JASP; University of Amsterdam, Amsterdam, The Netherlands). Statistical significance was set at *p* < 0.05.

## 3. Results

### 3.1. Characteristics of the Study Sample

The study sample was designed to be representative of the population, with demographic distributions reflecting the regional characteristics. Of the 4000 individuals invited, approximately 2120 returned the questionnaire, yielding a response rate of 53%. Among these, 2090 adults provided fully completed responses and were included in the analysis, corresponding to a completion rate of approximately 98.6% (Table 1).

The weighted gender distribution closely mirrors the population estimates. The weighted mean age was 53.8 years (SD: 17.6), with a median age of 54 years (IQR: 39–69 years). More than half of the respondents were in the oldest age group, while the youngest age group comprised the smallest proportion.

Most respondents were German speakers, followed by Italian speakers and other linguistic groups. The linguistic distribution in the weighted sample aligns with provincial census data, supporting representativeness. Most respondents held Italian citizenship. Most respondents resided in rural areas, whereas a smaller proportion lived in urban settings. Regarding living situations, most respondents lived with a partner or family, whereas a smaller proportion reported living alone. The educational levels varied, with a notable proportion having vocational training or a high school education, while a smaller segment had a university-level education.

Of the 2090 cases, 442 (21.1%) did not provide sufficient data to calculate a health literacy score. Among the remaining 1648 evaluable cases, health literacy was categorised as inadequate (266 cases, 16.1%), problematic (559 cases, 33.9%), or adequate (823 cases, 50.0%), based on the classification criteria described in the Methods Section.

### 3.2. Health Information Sources and Trust

Table 2 presents the descriptive statistics for the use of health information sources and trust in various health-related entities among the surveyed respondents. Regarding health information use, traditional media sources such as newspapers, magazines, television, and radio were used occasionally, but not as primary sources. Conversations with friends and acquaintances as well as direct discussions with healthcare professionals remained common sources of information, albeit with some variability. Formal educational events and the specialist literature were consulted less frequently, with the latter exhibiting broader variability in use.

Digital sources demonstrated mixed patterns. Targeted internet searches were employed more frequently than incidental exposure to health information online, which was slightly less prevalent. Social media and internet forums were the least frequently used sources of health information, as the majority of respondents reported using them rarely or never.

When considering trust in health information sources, medical professionals—including general practitioners, specialists, and nurses—were reported as the most trusted sources. However, trust scores were analysed separately from self-reported use, as defined in the Methods Section. Pharmacists were also considered reliable, although slightly less so than physicians. Books and personal experiences ranked higher than online sources and advice from friends and relatives. Conversely, trust in internet-based sources and social media was lower, reflecting limited perceived credibility of digital platforms for health information.

Overall, the findings confirmed a strong preference for obtaining health information directly from healthcare professionals, as compared to online or non-professional sources, emphasising the continued importance of trusted medical guidance in health-related decision-making.

Additional analyses provided in the Appendix A further detail the sociodemographic differences in health information use and trust. These analyses were conducted using non-weighted data. Women were more likely than men to use formal and interpersonal sources, whereas individuals with higher education used more digital and specialised sources. Linguistic differences were observed, with German speakers favouring traditional media and healthcare professionals, whereas Italian speakers exhibited a greater use of social media. In terms of trust, education level played a significant role, with individuals with a lower education level demonstrating greater reliance on family and the internet, while those with a higher education level placed more trust in books and professional sources. Age-related differences were also identified, with older individuals more likely to use traditional sources such as newspapers and TV, while younger individuals engaged more with digital platforms, particularly targeted internet searches and social media.

Appendix A illustrate the overall frequency distributions of health information use and trust, respectively, confirming the variation in reliance on professional, social, and digital sources.

### 3.3. Correlations of Health Activation, Health Literacy, and Health Status with Use of Information Sources

Weighted Spearman’s rank correlation coefficients were computed to explore associations between PAM-10 scores, HLS-EU-Q16 scores, and self-rated health status (0–100 scale), with the reported frequency of using health information sources (Table 3).

Results showed weak but consistent negative correlations between these scores and most health information sources. For example, PAM-10 scores correlated negatively with the use of healthcare professionals as information sources (ρ = −0.255, *p* < 0.001), indicating that respondents with higher activation levels reported a less frequent use of professional consultations for health information. HLS-EU-Q16 scores showed a similar negative correlation with the use of newspapers or magazines (ρ = −0.191, *p* < 0.001) and with social media (ρ = –0.172, *p* < 0.001).

Self-reported health status also showed negative correlations with the use of health information sources, though these were generally weaker. For instance, the correlation between health status and internet forums was ρ = −0.118 (*p* < 0.001), and with professional healthcare providers, ρ = −0.093 (*p* < 0.001).

Among all the sources, the strongest negative correlation for health status was observed with healthcare professionals, indicating that healthier individuals are less likely to report obtaining health information from doctors or nurses. Similar trends were found for friends and acquaintances and newspapers/magazines, suggesting a lower engagement with interpersonal and traditional media sources among those with better self-reported health. The weakest correlations were found for social media and internet forums.

### 3.4. Predictors of the Use of Health Information Sources

Ordinal regression models examined the association between demographic and health-related factors and the frequency of use of different health information sources. Across all models, AIC and BIC values indicated a reasonable model fit, with AIC values generally ranging between approximately 3600 and 4000 and BIC values ranging between 3700 and 4200. The deviance goodness-of-fit tests consistently returned *p*-values of 1.000, indicating no significant lack of fit, while Pearson goodness-of-fit tests demonstrated a better fit in some models compared to others, particularly online-related sources, where overdispersion was more apparent. Given the comparability of model performance and the importance of assessing the roles of all predefined predictors, the full models are presented in Table 4 to allow for a comprehensive evaluation of the demographic and health-related influences on health information use.

The results indicated that age, education, and health literacy were consistent predictors across multiple sources. Older individuals were more likely to rely on traditional media, such as newspapers and healthcare professionals, while younger individuals demonstrated a strong preference for online sources, including targeted internet searches, social media, and online forums. Education was positively associated with health information use across all sources, with higher education levels predicting greater engagement with the specialist literature, books, and online health searches. Gender differences were observed, with men being less likely to use newspapers and more likely to engage in social media and online forums. Language effects showed higher social media use among Italian speakers than German speakers, while the geographic region played a role in TV and radio use, which was more common in urban populations but had no significant effect on most other information sources.

Higher HLS-EU-Q16 scores were positively associated with the frequency of using various health information sources. The strongest correlations were observed with targeted internet searches (ρ = 0.225, *p* < 0.001), specialist literature (ρ = 0.203, *p* < 0.001), and healthcare professionals (ρ = 0.196, *p* < 0.001). Health status was negatively associated with the use of health information sources. For example, individuals reporting better health were less likely to use online forums (ρ = −0.118, *p* < 0.001), healthcare professionals (ρ = −0.093, *p* < 0.001), and newspapers or magazines (ρ = −0.084, *p* < 0.001).

### 3.5. Predictors of Trust in Sources of Health Information

Ordinal regression models examined the association between demographic and health-related factors and the level of trust in different health information sources. Across all models, AIC values ranged between approximately 2900 and 3700, whereas BIC values varied between 3000 and 3800, indicating a reasonable model fit. The deviance goodness-of-fit tests consistently returned *p*-values of 1.000, suggesting no significant lack of fit, whereas Pearson goodness-of-fit tests demonstrated superior fit in some models compared to others, particularly for trust in online sources, where overdispersion was more pronounced. Given the comparability of model performance and the theoretical significance of geographic and linguistic factors in South Tyrol, full models are presented to provide a comprehensive evaluation of demographic and health-related influences on trust in different health information sources.

Ordinal regression models identified key demographic and health-related variables influencing trust in various health information sources (Table 5). Health literacy (HLS-EU-Q16) was consistently associated with higher trust in traditional medical sources, such as family doctors, hospital specialists, and pharmacists, as well as in books and the literature. Patient activation (PAM-10) showed heterogeneous associations with trust in information sources. Higher PAM-10 scores were associated with a greater trust in nurses (OR = 1.23, 95% CI: 1.09–1.38, *p* = 0.001) and in personal experience (OR = 1.31, 95% CI: 1.17–1.46, *p* < 0.001), while no significant associations were observed for other sources such as the Internet, pharmacists, or books. Self-rated health was positively associated with trust in family doctors and books, suggesting that individuals with a higher self-reported health status tend to place greater confidence in these sources.

Age was positively associated with trust in traditional medical professionals, such as family doctors (OR = 1.34, 95% CI: 1.19–1.51, *p* < 0.001), and negatively associated with trust in Internet-based sources (OR = 0.82, 95% CI: 0.73–0.92, *p* = 0.001). Gender differences were limited, except for books and the literature, where males reported lower trust than females (OR = 0.87, 95% CI: 0.76–0.99, *p* = 0.035).

Educational attainment showed source-specific associations. Individuals with secondary education (compared to lower education) reported a higher trust in advice from family and friends (OR = 1.21, 95% CI: 1.02–1.44, *p* = 0.029), while tertiary education was associated with a greater trust in books and the literature (OR = 1.41, 95% CI: 1.20–1.66, *p* < 0.001).

Language group differences were particularly evident regarding trust in interpersonal sources: Italian speakers reported lower trust in friends and relatives compared to German speakers (OR = 0.75, 95% CI: 0.63–0.89, *p* = 0.001). No significant associations were found for geographic region (rural vs. urban).

## 4. Discussion

This study elucidates the essential roles of health literacy, education, and health-related factors in influencing both health information use and trust in health information sources. While healthcare professionals remained the most trusted sources, engagement with digital health information varied across population groups, with younger individuals and those with higher education demonstrating a greater reliance on digital sources. Health literacy emerged as an important factor associated with trust and the use of structured health information sources, such as the specialist literature and healthcare professionals. Lower educational levels were linked to a greater reliance on interpersonal sources, such as family and friends. Self-rated health status was positively associated with trust in structured and expert-based sources, including books and healthcare professionals. Patient activation showed positive associations with trust in healthcare professionals and educational sources; however, its relationship with digital information use was limited, suggesting that higher activation may support engagement with formal healthcare contexts, but not necessarily with online information. Linguistic and geographic effects were present but modest: Italian speakers were more likely to use social media than German speakers, and individuals living in urban areas reported a higher use of television and radio for health information. These findings reinforce the importance of improving health literacy, addressing disparities in information trust, and ensuring equitable access to reliable health information across different population groups.

### 4.1. The Role of Health Literacy and Patient Activation

This study reinforces the distinction between health literacy (HLS-EU-Q16) and patient activation (PAM-10) in shaping health information use and trust in health information sources. Health literacy primarily influences an individual’s ability to access, evaluate, and engage with structured health information, whereas patient activation is linked to motivation, confidence, and self-management behaviours in health contexts.

These findings align with Hibbard’s [30] framework, which conceptualises health literacy as a cognitive skill set for processing health information, whereas patient activation extends to confidence in managing one’s health and interacting with healthcare providers. Higher health literacy (HLS-EU-Q16) was consistently associated with greater use and trust in structured and professional sources, including books, the specialist literature, and healthcare professionals. These observations are in line with prior research highlighting the role of health literacy in navigating and evaluating evidence-based health information [31].

Patient activation (PAM-10), by contrast, showed a more selective pattern of associations. Higher activation scores were significantly associated with greater trust in nurses (OR = 1.23, *p* = 0.001) and personal experience (OR = 1.31, *p* < 0.001), but not with most other sources, including digital media. This suggests that while patient activation reflects confidence and engagement in managing one’s own health, it does not necessarily correspond to broader use or trust in publicly available health information sources outside of the formal healthcare context.

These distinctions reinforce the need to consider health literacy and patient activation as related but distinct dimensions when analysing health information behaviours.

The findings also highlight the importance of cultural and linguistic factors. Native language influenced health information use, particularly in online settings, with Italian speakers being more likely to trust social media than German speakers. The integration of linguistic and regional differences emphasises the need to consider sociocultural contexts when assessing the relationship between health literacy, activation, and trust in health information sources.

### 4.2. Age and Education

Age was a significant determinant of health information use. Older individuals demonstrated a greater propensity to rely on traditional media and healthcare professionals, consistent with previous studies indicating that older adults prefer familiar, authoritative sources over digital ones for health information [32,33,34]. This is partly due to heightened concerns about health risks [35]. Conversely, younger individuals reported higher engagement with social media and targeted internet searches, highlighting a generational shift towards digital health information use [36]. A 2014 investigation conducted in South Tyrol identified an ‘interpersonal’ health information-seeking group, characterised by a preference for information obtained through direct communication with friends and healthcare professionals [15]. This finding aligns with our study, where older adults reported the more frequent use of traditional media and professional sources, while younger individuals more often engaged with targeted internet searches and social media.

Education significantly influences both health information use and trust patterns [12,37]. Higher education was associated with greater engagement with the specialist literature and structured health resources, whereas lower education was linked to a greater reliance on family and informal sources. This supports the findings of Sørensen et al. [21], who indicate that education enhances health information use and fosters trust in evidence-based sources. Ausserhofer et al. [15] found that individuals with lower education were more likely to rely on interpersonal sources, whereas those with higher education engaged more frequently with digital and structured health sources. This is consistent with the present findings, showing that higher education levels were associated with a greater reliance on books, the professional literature, and healthcare professionals, whereas lower education was linked to an increased dependence on family and informal networks.

### 4.3. Trust in Health Information Sources

Consistent with previous research [12,38], healthcare professionals—particularly general practitioners and hospital specialists—were among the most trusted health information sources in this study. Trust in books and printed literature was moderate, while trust in internet sources was lower on average [39,40]. It has been suggested that this pattern reflects persistent scepticism towards the trustworthiness and accuracy of online health information. However, not all studies have confirmed this trend, particularly in contexts involving controversial or polarising health topics, where trust in professionals may be challenged, and alternative sources gain salience. Although healthcare professionals remain among the most trusted sources in many studies, the recent literature also highlights the increasing role of online sources in shaping health decisions, particularly in younger populations or among those who question professional authority. Trust in online health information is influenced by various design and content factors and does not always align with traditional credibility hierarchies [41]. Adolescents primarily rely on the internet for health information, often through social media and peer networks, rather than consulting online health professionals [42]. This contrasts with adult populations, where professional sources remain more trusted.

Notably, trust in family and personal experiences varied by education level, with individuals possessing lower educational attainment demonstrating greater reliance on interpersonal sources. This pattern is consistent with prior research, indicating that individuals with limited health literacy are more likely to use and trust non-professional sources such as family, friends, or social media, which may feel more accessible or relatable [12,43].

### 4.4. Linguistic and Regional Differences in Health Information-Seeking and Trust

Language plays a crucial role in shaping health information use beyond traditional sociodemographic factors, such as age and education. Ausserhofer et al. [15] identified distinct patterns of health information engagement among linguistic groups in South Tyrol, with German speakers more likely to rely on interpersonal and professional sources, while Italian speakers showed greater engagement with digital health sources. These findings agree with the results of the present study, where in the sample analysed, Italian speakers reported a higher reliance on social media for health information [44], whereas German speakers preferred traditional media and healthcare professionals. The observed differences may be due to cultural media consumption patterns [45,46], healthcare communication strategies, or varying levels of trust in public institutions across linguistic groups [47,48].

The findings, when juxtaposed with those of Ausserhofer et al. [15], elucidate the evolution of health information use in South Tyrol over time, particularly in the decade encompassing the pre- and post-pandemic periods. While both studies identified linguistic group differences in media preferences, the increased utilisation of digital sources, particularly among Italian speakers and younger individuals, became more pronounced in the years following the COVID-19 pandemic. This shift reflects broader trends in digital health information use, increased accessibility to online health information, and the growing influence of social media on shaping health perceptions. However, the sustained high level of trust in healthcare professionals across both studies suggests that traditional medical authorities remain the foundation of health information-seeking behaviours, even as digital sources gain prominence.

### 4.5. Strengths and Limitations

This study has several strengths. This is based on a representative cross-sectional survey conducted in South Tyrol, a region with a unique cultural and linguistic composition within an economically strong area of Central Europe. The stratified sampling design enabled representativeness, allowing for comparisons across age groups, linguistic backgrounds, and urban and rural populations. The inclusion of health literacy (HLS-EU-Q16), patient activation (PAM-10) and subjective health status along with sociodemographic variables provides a thorough analysis of the factors influencing health information use and trust in information sources. This study builds upon previous research from more than a decade ago, enabling the evaluation of changes over time, particularly in the post-pandemic period.

However, this study has several limitations that must be considered. As a cross-sectional study, causal relationships cannot be established between predictors and health information use or trust. Self-reported data introduce potential recall bias and social desirability effects, particularly in responses related to trust in sources and digital health information use. While stratified sampling ensured representativeness, non-response bias could not be fully excluded. This study relied on subjective self-assessments of health literacy and patient activation, which, while validated, may differ from objective measures.

The survey did not differentiate between interpersonal and media-based engagement with healthcare professionals as information sources. Thus, interpretations regarding the role of health literacy in shaping direct versus indirect information-seeking behaviours should be made with caution.

From a statistical perspective, collinearity between predictors has not been formally assessed, although theoretical considerations suggest that multicollinearity is unlikely to have a substantial impact on interpretability because of the low correlation coefficients observed (Table 3). Although the sample size was sufficient to support ordinal regression modelling, the number of variables and categorical predictors could contribute to model complexity. While AIC, BIC, and goodness-of-fit tests indicated robust model performance, potential interactions or collinearity among predictors may influence individual parameter estimates. Future research could further explore these relationships with alternative modelling approaches. Future research may benefit from variance inflation factor (VIF) analysis or structural equation modelling approaches to further explore complex interrelationships. While ordinal regression is an appropriate methodological choice, given the ordered nature of dependent variables, alternative modelling approaches could be explored in future studies to refine predictions and enhance interpretability. Although multiple tests were performed, this study follows an exploratory approach, and findings should be interpreted in context rather than based solely on *p*-values. While adjusting for multiple comparisons (e.g., Bonferroni correction) can reduce the risk of Type I errors, it may also increase Type II errors, potentially masking relevant associations. Therefore, significance levels should be considered alongside effect sizes and theoretical justification rather than strictly corrected.

Notwithstanding these constraints, this study offers insightful observations on how individuals seek health information and their confidence in various health sources across a heterogeneous European region.

### 4.6. Implications for Public Health Strategies

The independent but complementary roles of health literacy and patient activation suggest that interventions should be tailored accordingly. (1) Health literacy initiatives should focus on enhancing individuals’ abilities to access, evaluate, and utilise evidence-based health information, particularly in online environments where misinformation is prevalent. (2) Patient activation strategies should emphasise confidence-building measures to enhance patient–provider communication and engagement in formal healthcare settings.

Moreover, the findings highlight the need for future research on digital health literacy, particularly on how it intersects with activation and trust in online sources. The further exploration of linguistic and regional disparities in health information use could provide a better understanding of how health literacy and patient activation function across diverse populations.

Findings from this study and Ausserhofer et al. [15] reinforce the need for demographically targeted health communication strategies, particularly in linguistically and culturally diverse regions, such as South Tyrol.

Linguistic differences: Health information campaigns should ensure equitable access to accurate and reliable health information across German and Italian-speaking populations.Education-based disparities: Efforts should focus on simplifying complex health information for lower-educated populations, while providing advanced resources for highly educated individuals.Age-related digital engagement: Strategies should emphasise bridging the digital divide for older adults while ensuring that younger individuals critically evaluate online health sources.

By integrating these insights, sociodemographic influences on health information use and trust can be better understood, strengthening the case for tailored, linguistic, and culturally adaptive health communication strategies in South Tyrol.

## 5. Conclusions

This study analyses health information use and trust in health information sources in South Tyrol, a diverse region of Central Europe. The findings highlight the role of health literacy, education, and demographics in shaping engagement with and trust in health information sources. Healthcare professionals remain the most trusted, but younger individuals and Italian speakers rely more on digital sources, whereas older individuals and German speakers trust traditional media and interpersonal sources more.

This study emphasises the independent roles of health literacy and patient activation. Higher health literacy is associated with engagement in structured and professional health sources, while higher patient activation predicts greater trust in healthcare professionals.

Despite its cross-sectional design and dependence on self-reported data, this study offers insights into evolving health information use patterns, particularly after the pandemic. A comparison with previous research suggests a shift towards digital sources, but with persistent disparities in information trust across demographic groups. Public health strategies should focus on ensuring equitable access to accurate health information, addressing digital literacy disparities, and integrating trusted health professionals into digital health communication.

Future research should explore the role of social media in health communication, the long-term effects of digital health literacy interventions, and the interplay between trust, misinformation, and engagement with health services. South Tyrol serves as a valuable case study for understanding how sociodemographic factors shape health information seeking and trust in broader European and global contexts.

## Figures and Tables

**Figure 1 ijerph-22-00570-f001:**
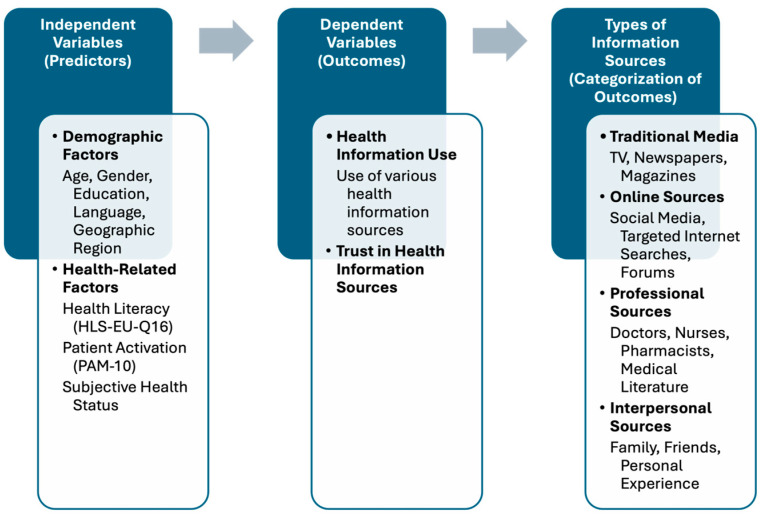
Conceptual framework of predictors and outcomes in the use of health information and trust in information sources.

**Table 1 ijerph-22-00570-t001:** Demographic and other characteristics of the sample were the weighted distribution of categorical variables, including sociodemographic characteristics, health literacy (HLS-EU-Q16), patient activation (PAM-10), and self-reported health status (*n* = 2090).

Variable	Category	Weighted *n*	Weighted Proportion %	95% CI Low %	95% CI High %
Gender	Female	1158	55.5	54.2	56.8
	Male	932	44.5	43.2	45.8
Age Group (Years) ^1^	18–34	378	18.1	16.9	19.3
	35–54	643	30.7	29.3	32.1
	55+	1070	51.2	49.7	52.7
Native Language	German	1395	66.8	65.5	68.1
	Italian	499	23.9	22.7	25.1
	Other	194	9.3	8.5	10.1
Citizenship	Italian	2009	96.1	95.4	96.8
	Other	81	3.9	3.2	4.6
Community Type	Urban	387	18.5	17.3	19.7
	Rural	1703	81.5	80.3	82.7
Living Situation	Alone	378	18.1	17.0	19.3
	With partner/family	1327	63.5	62.1	64.9
	With children	787	37.7	36.3	39.1
Educational Level	Middle school or lower	487	23.3	22.0	24.6
	Vocational school	670	32.1	30.7	33.5
	High school	533	25.6	24.3	26.9
	University	396	19.0	17.9	20.1
Health Literacy ^2^	Inadequate	266	16.1	14.2	18.0
	Problematic	559	33.9	31.5	36.3
	Adequate	824	50.0	47.4	52.5
PAM-10 ^3^	Disengaged	340	16.3	15.3	17.3
	Becoming aware	888	42.6	41.2	44.0
	Taking action	648	31.1	29.8	32.4
	Maintaining	204	9.8	9.0	10.6
Health Status(0–100) ^4^	Poor (0–50)	255	12.2	11.3	13.1
	Fair (51–75)	625	29.8	28.5	31.1
	Good (76–90)	820	39.1	37.8	40.4
	Excellent (91–100)	390	18.9	17.9	19.9

^1^ weighted mean age of 53.8 years (standard deviation of 17.6); median of 54 years (interquartile range of 39–69). ^2^ HLS-EU-Q16: health literacy was grouped into inadequate (0–25), problematic (26–33), and adequate (34–50). ^3^ PAM-10: patient activation was categorised into four levels: low (≤47.0, disengaged) to high (≥72.5, maintaining behaviours). ^4^ Health status: categorised into poor (0–50), fair (51–75), good (76–90), and excellent (91–100). CI, confidence interval.

**Table 2 ijerph-22-00570-t002:** Sources of and trust in health information (*n* = 2090).

Item ^1^	Median	IQR	Mean	SD
How do you generally engage with health information?				
Conversations with friends or acquaintances	2	1	2.35	0.81
Discussions with specialists, e.g., doctors or nursing staff	2	1	2.40	0.89
Targeted internet searches for health information	2	1	2.50	1.01
Television or radio programmes on health topics	3	1	2.65	0.91
Articles in newspapers or magazines	3	1	2.69	0.90
Incidental exposure to health information online (e.g., while browsing)	3	2	2.80	0.93
Specialist literature, e.g., health encyclopaedias or how-to books	3	2	3.10	0.94
Social networks (Facebook, Instagram, etc.)	4	1	3.28	0.93
Events or courses	4	1	3.32	0.86
In internet forums in which personal questions are asked or answered	4	1	3.33	0.89
How much do you trust the following sources for health information?				
The specialists in the outpatient clinics or hospitals	2	1	1.70	0.68
Your family doctor	2	1	1.72	0.72
Your own feeling or experience	2	1	1.85	0.64
The pharmacists	2	0	1.95	0.65
The nurses	2	0	1.97	0.69
Information from books	2	1	2.38	0.83
The advice of friends or relatives	3	1	2.56	0.69
Health information from the Internet	3	1	2.97	0.74

^1^ weighted descriptive statistics of health information use and trust ratings: 1 = regularly or very, 2 = occasionally or quite, 3 = rarely or little, and 4 = never or not at all. Abbreviations: SD, standard deviation; IQR, interquartile range.

**Table 3 ijerph-22-00570-t003:** Correlation between patient activation, health literacy, and the use of health information sources.

**Information Source**	** *n* **	**Patient Activation** **(PAM-10)**		**Health Literacy** **(HLS-EU-Q16)**		**Health Status** **(0–100 Scale)**	
		ρ	** *p* ** **-Value**	ρ	** *p* ** **-Value**	ρ	** *p* ** **-Value**
Newspapers/Magazines	2079	−0.145	<0.001	−0.191	<0.001	−0.127	<0.001
TV/Radio	2079	−0.083	<0.001	−0.111	<0.001	−0.094	<0.001
Friends/Acquaintances	2079	−0.157	<0.001	−0.121	<0.001	−0.134	<0.001
Healthcare Professionals	2079	−0.255	<0.001	−0.167	<0.001	−0.201	<0.001
Events/Courses	2079	−0.176	<0.001	−0.133	<0.001	−0.116	<0.001
Specialist Literature	2079	−0.143	<0.001	−0.143	<0.001	−0.099	<0.001
Incidental Exposure to Health Information Online	2079	−0.097	<0.001	−0.097	<0.001	−0.076	<0.001
Targeted Internet Search	2079	−0.130	<0.001	−0.130	<0.001	−0.090	<0.001
Internet Forums	2079	−0.071	<0.001	−0.071	<0.001	−0.053	<0.001
Social Media	2079	−0.058	<0.001	−0.058	<0.001	−0.044	<0.001

ρ, Spearman’s rank correlation coefficient.

**Table 4 ijerph-22-00570-t004:** Generalised linear regression model output for predictors of the use of health information sources.

Information Source	Regression Coefficient β, Odds Ratio [95% Confidence Interval], *p*-Value ^1^								
	Intercept	HLS-EU-Q16	PAM-10	Health Status	Age	Gender ^2^	Education ^3^	Language ^4^	Rural/Urban ^5^
Mass Media									
Newspapers/Magazines	−8.012 [−9.920; −6.104] <0.001	0.205 [0.125; 0.284] <0.001	0.016 [0.000; 0.033] n.s.	0.003 [−0.010; 0.016] n.s.	0.057 [0.042; 0.072] <0.001	−1.037 [−1.513; −0.561] <0.001	1.000 ^6^ [0.612; 1.387] <0.001	−0.140 [−0.515; 0.235] n.s.	0.431 [−0.217; 1.043] n.s.
TV/Radio	−5.924 [−7.706; −4.142] <0.001	0.105 [0.036; 0.174] 0.003	0.013 [−0.004; 0.029] n.s.	0.003 [−0.010; 0.015] n.s.	0.050 [0.036; 0.064] <0.001	−0.970 [−1.407; −0.533] <0.001	−0.316 ^6^ [−0.676; 0.045] n.s.	−0.089 [−0.476; 0.279] n.s.	0.818 [0.181; 1.455] 0.012
Personal Contacts									
Friends/Acquaintances	1.115 [−0.707; 2.937] n.s.	0.109 [0.036; 0.182] 0.004	0.008 [−0.011; 0.027] n.s.	0.016 ^7^ [0.004; 0.028] 0.007	−0.037 [−0.052; −0.022] <0.001	−1.215 [−1.700; −0.730] <0.001	0.274 ^8^ [−0.260; 0.808] n.s.	0.199 [−0.343; 0.741] n.s.	−0.029 [−0.694; 0.636] n.s.
Healthcare Professionals	−3.132 [−4.714; −1.550] <0.001	0.134 [0.070; 0.197] <0.001	0.036 [0.019; 0.053] <0.001	−0.023 [−0.034; −0.011] <0.001	0.011 [−0.001; 0.024] n.s.	−0.034 [−0.437; 0.368] n.s.	1.449 ^6^ [0.870;2.028] <0.001	0.126 [−0.347; 0.598] n.s.	0.422 [−0.139; 0.984] n.s.
Educational and Academic Sources									
Events/Courses	−4.567 [−6.694; −2.439] <0.001	0.073 [−0.025; 0.171] n.s.	0.044 [0.025; 0.063] <0.001	−0.007 [−0.023; 0.009] n.s.	−0.026 [−0.044; −0.008] 0.005	−0.805 [−1.376; −0.234] 0.006	1.449 ^6^ [0.870; 2.028] <0.001	0.046 [−0.327; 0.418] n.s.	0.063 [−0.669; 0.796] n.s.
Specialist Literature	−5.971 [−7.712; −4.229] <0.001	0.061 [−0.012; 0.134] n.s.	0.044 [0.029; 0.060] <0.001	−0.007 [−0.020; 0.006] n.s.	0.017 [0.003; 0.031] 0.015	−0.774 [−1.220; −0.328] <0.001	1.051 ^6^ [0.693; 1.410] <0.001	−0.579 [0.936; 0.221] <0.001	0.237 [−0.374; 0.848] n.s.
Online Sources									
Incidental Exposure to Health Information Online	0.049 [−1.645; 1.744] n.s.	0.111 [0.038; 0.184] 0.003	0.005 [−0.012; 0.022] n.s.	0.000 [−0.013; 0.012] n.s.	−0.060 [−0.074; -0.046] <0.001	−0.239 [−0.678; 0.199] n.s.	0.442 ^6^ [0.071; 0.813] 0.020	0.380 [0.022; 0.738] 0.037	0.002 [−0.603; 0.607] n.s.
Targeted internet Search	1.550 [0.033; 3.066] 0.045	0.083 [0.026; 0.141] 0.005	0.009 [−0.006; 0.024] n.s.	0.002 [−0.008; 0.013] n.s.	−0.070 [−0.082; −0.057] <0.001	−0.265 [−0.628; 0.098] n.s.	1.479 ^6^ [0.902; 2.055] <0.001	0.386 [−0.031; 0.802] n.s.	0.211 [−0.335; 0.757] n.s.
Internet Forums	−2.071 [−4.182; 0.040] n.s.	0.084 [−0.006; 0.173] n.s.	0.000 [−0.020; 0.021] n.s.	−0.003 [−0.020; 0.014] n.s.	−0.032 [−0.049; −0.015] <0.001	−0.535 [−1.079; 0.008] n.s.	0.579 ^6^ [−0.088;1.246] n.s.	0.300 [−0.056; 0.656] n.s.	0.124 [−0.610; 0.857] n.s.
Social Media	−0.095 [−1.907; 1.718] n.s.	0.105 [0.024; 0.186] 0.011	0.002 [−0.016; 0.020] n.s.	0.000 [−0.015; 0.013] n.s.	−0.067 [−0.083; −0.052] <0.001	−0.796 [−1.279; −0.312] 0.001	−0.355 ^6^ [−0.950; 0.241] n.s.	0.640 ^9^ [0.288; 0.992] <0.001	−0.008 [−0.667; 0.651] n.s.

^1^ unless otherwise indicated, regression coefficients represent the effect of the predictor on the cumulative log-odds of reporting a more frequent use of information sources (e.g., ‘Seldom’, ‘Sometimes’, or ‘Regularly’) compared to all lower categories (e.g., ‘Never’). The proportional odds model assumes that this relationship holds across all thresholds of the ordinal outcome. ^2^ reference category: female. ^3^ education level: primary (middle school), secondary (high school), or tertiary (university). ^4^ reference category: German. Coefficients for ‘Italian’ and ‘Other’ represent comparisons to German speakers. Language was treated as a nominal variable with three categories (German, Italian, and Other). ^5^ reference category: rural. ^6^ effect of a tertiary education level on the log-odds of reporting “Sometimes” or “Often” rather than “Rarely” or “Never”, compared to the primary education level. ^7^ effect of better self-rated health on the log-odds of reporting “Sometimes” or “Often” rather than “Rarely” or “Never”, compared to individuals with lower self-rated health. ^8^ effect of secondary education level on the log-odds of reporting “Sometimes” or “Often” rather than “Rarely” or “Never”, compared to the primary education level. ^9^ effect of speaking Italian on the log-odds of reporting ‘Sometimes’ or ‘Often’ rather than ‘Rarely’ or ‘Never’, compared to the reference category of German.

**Table 5 ijerph-22-00570-t005:** Generalised linear regression model outputs for predictors of trust in sources of health information.

Information Source	Regression Coefficient β, Odds Ratio [95% Confidence Interval], *p*-Value ^1^								
	Intercept	HLS-EU-Q16	PAM-10	Health Status	Age	Gender ^2^	Education ^3^	Language ^4^	Rural/Urban ^5^
Healthcare Professionals									
Family Doctor	−3.628 [−6.458; −0.798] 0.012	0.284 [0.174; 0.393] <0.001	−0.005 [−0.039; 0.029] n.s.	0.033 [0.016; 0.050] <0.001	0.034 [0.009; 0.059] 0.007	−0.272 [−1.095; 0.551] n.s.	0.745 ^6^ [−0.465;1.954] n.s.	−0.233 [−1.326; 0.860] n.s.	0.228 [−0.913; 1.369] n.s.
Hospital Specialists	−2.651 [−5.888; 0.585] n.s.	0.144 [0.023; 0.265] 0.019	0.033 [−0.008; 0.074] n.s.	0.013 [−0.008; 0.034] n.s.	0.031 [0.004; 0.058] 0.023	−0.144 [−1.016; 0.728] n.s.	1.181 ^6^ [−0.130;2.492] n.s.	0.352 [−1.039; 1.744] n.s.	−0.279 [−1.648; 1.089] n.s.
Pharmacists	0.584 [−2.914; 4.083] n.s.	0.193 [0.065; 0.322] 0.003	0.007 [−0.033; 0.047] n.s.	0.005 [−0.018; 0.028] n.s.	−0.016 [−0.045; 0.014] n.s.	−0.886 [−1.537; 0.032] n.s.	1.181 ^6^ [−0.130; 2.492] n.s.	0.773 [−0.618; 2.164] n.s.	0.581 [−0.691; 1.852] n.s.
Nurses	0.922 [−1.942; 3.785] n.s.	0.055 [−0.052; 0.161] n.s.	0.040 [0.007; 0.073] 0.017	0.000 [−0.020; 0.019] n.s.	−0.020 [−0.042; 0.003] n.s.	−0.298 [−0.993; 0.398] n.s.	0.911 ^6^ [−0.185; 2.008] n.s.	0.969 [−0.197; 2.136] n.s.	−0.527 [−1.699; 0.645] n.s.
Personal and Social Trust									
Personal Experience	−2.698 [−5.924; 0.528] n.s.	0.116 [0.013; 0.244] n.s.	0.052 [0.009; 0.095] 0.017	0.026 [0.007; 0.046] 0.008	0.009 [−0.019; 0.038] n.s.	−1.214 [−2.217; −0.210] 0.018	0.721 ^6^ [−0.618; 2.059] n.s.	−0.188 [−1.519; 1.143] n.s.	0.293 [−0.965; 1.552] n.s.
Family and Friends	−0.394 [−3.018; 2.231] n.s.	0.116 ^7^ [0.051; 0.181] <0.001	0.018 [−0.008; 0.044] n.s.	0.000 [−0.019; 0.018] n.s.	−0.019 [−0.040; 0.002] n.s.	0.031 [−0.622; 0.684] n.s.	1.032 ^6^ [0.367;1.698] 0.002	−1.125 [−1.663; −0.586] <0.001	−0.323 [−1.349; 0.704] n.s.
Educational and Media Sources									
Books and Literature	−2.318 [−4.120; −0.517] 0.012	0.108 [0.038; 0.177] 0.002	0.022 [0.003; 0.041] 0.021	0.017 [0.004; 0.030] 0.010	−0.032 [−0.046; −0.018] <0.001	−0.504 [−0.951; −0.057] 0.027	1.522 [0.833; 2.211] <0.001 ^8^	0.095 [−0.395; 0.505] n.s.	0.228 [−0.418; 0.874] n.s.
Internet Information	−2.514 [−5.817; 0.790] n.s.	0.136 [0.000; 0.271] 0.049	0.000 [−0.020; 0.006] n.s.	0.008 [−0.007; 0.011] n.s.	−0.038 [−0.063; −0.013] 0.003	−0.010 [−0.787; 0.766] n.s.	0.593 ^5^ [0.241; 0.945] <0.001	−0.313 [−1.428; 0.803] n.s.	−0.038 [−1.204: 1.127] n.s.

^1^ unless otherwise indicated, regression coefficients represent the effect of the predictor on the cumulative log-odds of reporting a more frequent use of information sources (e.g., ‘Seldom’, ‘Sometimes’, or ‘Regularly’) compared to all lower categories (e.g., ‘Never’). The proportional odds model assumes that this relationship holds across all thresholds of the ordinal outcome. ^2^ reference category: female. ^3^ education level: primary (middle school), secondary (high school), or tertiary (university). ^4^ reference category: German. Coefficients for ‘Italian’ and ‘Other’ represent comparisons to German speakers. Language was treated as a nominal variable with three categories (German, Italian, and Other). ^5^ reference category: rural. ^6^ effect of a secondary education level on the log-odds of reporting ‘Sometimes’ or ‘Often’ rather than ‘Rarely’ or ‘Never’, compared to the primary education level. ^7^ effect of higher health literacy on the log-odds of reporting ‘Sometimes’ or ‘Often’ rather than ’Rarely’ or ‘Never’, compared to individuals with lower health literacy. ^8^ effect of a university degree (tertiary level) on the log-odds of reporting ‘Sometimes’ or ‘Often’ rather than ‘Rarely’ or ‘Never’, compared to the primary education level.

## Data Availability

Data are available from the corresponding author upon reasonable request.

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
