# Peer review of "Health Information Use and Trust: The Role of Health Literacy and Patient Activation in a Multilingual European Region"

_ijerph, 2025, doi:10.3390/ijerph22040570_

Round 1
Reviewer 1 Report
Comments and Suggestions for Authors
Abstract - The abstract is written with clarity.
Introduction—The introduction section provides a high-level introduction to the research subject. It is suggested that it be enhanced by references to current healthcare problems and issues.
Data Preparation & Data Selection - It is recommended that these two sections be combined into the Methods section for clarity.
4.3. Normalization and integrated dataset - is used: 𝑍= (𝑥−? )/ 𝜎 (1)
Where:
X: Variable to transform.
μ: Average value of all samples of the variable to be transformed - This variable is missing in the equation. Please clarify.
σ: Standard deviation of all samples of the variable to be transformed.
Unified Dateset - It is advisable to enhance this by including the research outcomes; for instance, what aspects of the equation contribute to rehabilitation improvement?
Reviewer 2 Report
Comments and Suggestions for Authors
Comments.
The study is interesting and addresses important aspects such as health information, literacy and origin of the information.
The methodology used is valid and accounts for the analyses presented. The sample size is adequate, so it reduces the limitations in the use of ordinal regression models. However, the number of variables and categories could influence the results. I suggest adding some lines regarding this possible limitation, if it was considered or did not influence the results.
An additional comment is that the other is that adolescents usually their main source of information is the Internet and not necessarily online health professionals. Add a comment regarding this comparison.
Author Response
Comment 1: Influence of Variables and Categories in Ordinal Regression Models:
Response 1: We have added a statement in the Limitations section addressing how the number of variables and categories in the models could influence results. While the sample size mitigates potential biases, we acknowledge that model complexity could impact parameter stability. Inserted text:
“Although the sample size was sufficient to support ordinal regression modeling, the number of variables and categorical predictors could contribute to model complexity. While AIC, BIC, and goodness-of-fit tests indicated robust model performance, potential interactions or collinearity among predictors may influence individual parameter estimates.”
Comment 2: Comparison with Adolescents' Health Information-Seeking Behavior:
Response 2: The following note in the Discussion section was added:
“Adolescents primarily rely on the Internet for health information, often through social media and peer networks, rather than consulting online health professionals. This contrasts with adult populations, where professional sources remain more trusted.“
Reviewer 3 Report
Comments and Suggestions for Authors
Detailed comments can be found in the attached file. Thank you.

Terminology revisions are required, as detailed in the attached file.
Round 2
Reviewer 3 Report
Comments and Suggestions for Authors
Please see the attached.

The writing requires improvement (see the attached). Generative AI often introduces inconsistencies or errors (at times masked by superfluous language) and can dilute the precision of what the authors intend to convey.
Author Response
We sincerely thank the reviewer for the careful, constructive, and insightful comments provided throughout the manuscript. Your feedback greatly contributed to improving the clarity, conceptual precision, and methodological consistency of our work. We particularly appreciate the detailed attention to terminology (e.g., distinguishing between trust and credibility, use and reliance), the encouragement to include more precise and data-driven interpretations, and the suggestions to avoid overgeneralisation of results.
In the following responses, we have addressed each point thoroughly and revised the manuscript accordingly. We believe that the revised version now better reflects the complexity of the findings and aligns more consistently with both the data and the broader literature. Once again, thank you for your excellent support in strengthening this manuscript.
